# Immunogenicity of a Third Dose of BNT162b2 Vaccine among Lung Transplant Recipients—A Prospective Cohort Study

**DOI:** 10.3390/vaccines11040799

**Published:** 2023-04-04

**Authors:** Yael Shostak, Mordechai R. Kramer, Omer Edni, Ahinoam Glusman Bendersky, Noa Shafran, Ilana Bakal, Moshe Heching, Dror Rosengarten, Dorit Shitenberg, Shay M. Amor, Haim Ben Zvi, Barak Pertzov, Hila Cohen, Shahar Rotem, Uri Elia, Theodor Chitlaru, Noam Erez, Yuri Peysakhovich, Yaron D. Barac, Amir Shlomai, Erez Bar-Haim, Osnat Shtraichman

**Affiliations:** 1Department of Medicine D, Beilinson Hospital, Petah Tikva 4941492, Israel; edniomer@gmail.com (O.E.); achinoamg@gmail.com (A.G.B.); noashafran@gmail.com (N.S.); shlomaiamir@gmail.com (A.S.); 2Pulmonary Institute, Rabin Medical Center, Petach Tikva 4941492, Israel; kremerm@clalit.org.il (M.R.K.); ilanab2@clalit.org.il (I.B.); moshehe@clalit.org.il (M.H.); drorros@clalit.org.il (D.R.); dorits2@clalit.org.il (D.S.); mosheshaiam@clalit.org.il (S.M.A.); pertzovb@gmail.com (B.P.); osnatlivne@hotmail.com (O.S.); 3Sackler Faculty of Medicine, Tel Aviv University, Tel Aviv 6997801, Israel; 4Clinical Microbiology Laboratory, Beilinson Hospital, Petah Tikva 4941492, Israel; haimbe@clalit.org.il; 5Department of Biochemistry and Molecular Genetics, Israel Institute for Biological Research, Ness Ziona 7410001, Israel; hilac@iibr.gov.il (H.C.); shaharr@iibr.gov.il (S.R.); urie@iibr.gov.il (U.E.); theoc@iibr.gov.il (T.C.); erezb@iibr.gov.il (E.B.-H.); 6Department of Infectious Diseases, Israel Institute for Biological Research, Ness Ziona 7410001, Israel; noame@iibr.gov.il; 7Cardiothoracic Surgery Department, Rabin Medical Center, Petach Tikva 4941492, Israel; yuripe1976@yahoo.com (Y.P.); yaronbar@icloud.com (Y.D.B.)

**Keywords:** lung transplantation, BNT162b2 COVID-19 vaccine, third dose, immunogenicity

## Abstract

Two doses of mRNA SARS-CoV-2 vaccines elicit an attenuated humoral immune response among immunocompromised patients. Our study aimed to assess the immunogenicity of a third dose of the BNT162b2 vaccine among lung transplant recipients (LTRs). We prospectively evaluated the humoral response by measuring anti-spike SARS-CoV-2 and neutralizing antibodies in 139 vaccinated LTRs ~4–6 weeks following the third vaccine dose. The t-cell response was evaluated by IFNγ assay. The primary outcome was the seropositivity rate following the third vaccine dose. Secondary outcomes included: positive neutralizing antibody and cellular immune response rate, adverse events, and COVID-19 infections. Results were compared to a control group of 41 healthcare workers. Among LTRs, 42.4% had a seropositive antibody titer, and 17.2% had a positive t-cell response. Seropositivity was associated with younger age (t = 3.736, *p* < 0.001), higher GFR (t = 2.355, *p* = 0.011), and longer duration from transplantation (t = −1.992, *p* = 0.024). Antibody titer positively correlated with neutralizing antibodies (r = 0.955, *p* < 0.001). The current study may suggest the enhancement of immunogenicity by using booster doses. Since monoclonal antibodies have limited effectiveness against prevalent sub-variants and LTRs are prone to severe COVID-19 morbidity, vaccination remains crucial for this vulnerable population.

## 1. Introduction

Coronavirus disease 2019 (COVID-19) has contributed to a substantial mortality burden globally, attributed to over 6.5 million deaths through January 2023 [1]. Vaccine administration, advances in medical management, antiviral medications, and monoclonal antibodies have significantly decreased the morbidity and mortality associated with COVID-19. However, the pandemic is ongoing, with over 400,000 new cases diagnosed daily worldwide, allowing for the continued introduction of mutations in the viral genome resulting in increasing viral immune escape from available interventions [2]. In addition, the use of monoclonal antibodies, which were initially considered a vital tool in the fight against COVID-19 for at-risk populations [3], is now being challenged due to their lack of effectiveness against the newly emerged sub-variants. As a result, official health authorities were forced to reassess their approach, leading them to withdraw from this form of pre-exposure prophylaxis protection among vulnerable populations [4]. This has further emphasized the need for ongoing research on vaccine efficacy, the value of booster doses, and the development of variant-specific boosters. 

Standard regimens of COVID-19 mRNA vaccines are highly immunogenic in immunocompetent individuals [5,6]. Although two doses of mRNA BNT162b2 vaccines provided limited protection against symptomatic COVID-19 caused by the Omicron variant, the BNT162b2 third (booster) dose significantly increased protection. This protection waned over time [7], underscoring the importance of repeated variant-specific booster doses.

Solid organ transplant recipients (SOTR) are at increased risk for severe COVID-19 disease, hospitalization, and death [8,9], particularly lung transplant recipients (LTRs) [10,11]. The standard two-dose regimen of mRNA-based SARS-CoV-2 vaccines has elicited attenuated humoral and cellular immune responses among SOTR [12,13]. 

We have previously shown that only 18% of LTRs developed a positive titer to the spike protein (S-IgG) of severe acute respiratory syndrome coronavirus 2 (SARS-CoV-2) measured within three weeks of receiving the second mRNA BNT162b2 vaccine dose [14]. In the same study cohort, only 9.3% developed an INFγ cellular response (data not published). Low antibody responses in this population correlate with older age, the shorter time elapsed from transplantation, and concurrent antimetabolite treatment. Although improvement in antibody titers following a third vaccine dose has been previously reported in SOTR [15,16,17], there is a paucity of data regarding humoral and cellular response in LTRs following a third vaccine dose.

Given the heightened risk for severe COVID-19 disease among LTRs and data showing viral immune escape from monoclonal antibodies, our study aimed to evaluate the humoral and cellular response of a third dose of the BNT162b2 vaccine among LTRs. We hypothesized that we would observe an increase in immunogenicity among LTRs following a third vaccine dose, similar to data following the second dose.

## 2. Materials and Methods

### 2.1. Study Population

We conducted a prospective cohort study at Rabin Medical Center of lung and heart-lung transplant recipients who received the BNT162b2 mRNA vaccine between December 2020 and December 2021. Inclusion criteria were adult (≥18 years) lung and heart-lung transplant recipients who received three doses of the BNT162b2 vaccine. Patients were vaccinated as advised by the Israel Ministry of health at three time points: the first dose between December 2020 and January 2021, and the second dose was administrated 3 weeks following the first dose. The third dose was available for eligible at-risk populations as advised by the Israel Ministry of health between July 2021 and august 2021. Exclusion criteria were LTRs within 30 days post-transplantation and LTRs previously diagnosed with COVID-19 (as documented by a positive PCR nasal swab). The control group included immunocompetent healthcare workers. All participants provided written informed consent. The institutional review board approved the study (IRB-1069-20).

### 2.2. Data Collection

LTRs were evaluated, and whole blood samples were drawn for antibody titers, leukocyte and lymphocyte count, creatinine level, and immunosuppressive drug trough levels (calcineurin and mTOR inhibitors), 4–6 weeks and then again 10–15 weeks following the third vaccine dose. These were compared to baseline antibody titers drawn 9–18 weeks before the third vaccine dose. Whole blood samples for antibody titer in the control group were drawn nine weeks before and 3–5 months following the third vaccine dose. Relevant demographic and clinical data were derived from the electronic medical records at the Rabin Medical Center, including immunosuppressive drug regimens. Our standard immunosuppressive regimen includes calcineurin inhibitors (CNI), antimetabolite (most commonly mycophenolate mofetil-based), and steroids. In addition, an mTOR Inhibitor (everolimus) is used, in our practice, usually as an add-on renal-sparing agent combined with low-dose CNI. Comorbidities, such as ischemic heart disease (IHD), diabetes mellitus (DM), and congestive heart failure (CHF), were defined for each participant if they appeared in the medical record or if the subject received disease-related drug therapy. Glomerular filtration rate (GFR) was calculated using the CKD-EPI equation for each study participant at the beginning of the study. The immunosuppressive variable was calculated as the mean CNI drug trough levels or CNI and mTOR combined drug trough levels in the three months before entering the study. For patients treated with cyclosporin A, the trough blood levels were divided by 20 to integrate these data in the CNI trough levels. 

### 2.3. Humoral Response

The procedure involved using a centrifuge to separate serum from whole blood samples. The serum was then divided into smaller portions and frozen until the serological assay could be performed. The ARCHITECT^®^ i2000SR immunoassay analyzer was utilized to perform the SARS-CoV-2 IgG II Quant assay from the Abbott Ireland Diagnostic Division in accordance with the manufacturer’s package insert. To quantitatively determine the amount of immunoglobulin class G (IgG) antibodies to the receptor binding domain (RBD) of the S1 subunit of the spike protein of SARS-CoV-2 (S-IgG) in human serum and plasma samples, the chemiluminescent microparticle immunoassay (CMIA) was implemented. The method entails coating paramagnetic micro-particles with the antigen, which binds to S-IgG antibodies, following which an anti-human IgG acridinium-labeled conjugate is added. The resulting reaction creates a chemiluminescent response that measures the relative light unit (RLU). The detected RLU is directly proportional to the S-IgG in the sample, with S-IgG titers of 50 AU/mL and above being considered positive in the immunoassay test.

Antibody neutralization assays were performed as previously described [18]. Briefly, hACE2-expressing HEK293 cells were plated in a white-wall 96-well plate (2 × 104 cells per plate). One day later, on the assay day, heat-inactivated sera were serially diluted and mixed with pseudovirus expressing SARS-CoV-2 spike protein. Twenty-four hours later, cells were lysed, and luciferase activity (in relative light units (RLU)) was measured. Percent neutralization was normalized considering uninfected cells as 100% neutralization and cells infected with only pseudovirus as 0% neutralization. IC50 titers were determined using a log (agonist) vs. normalized response (variable slope) nonlinear function using Prism software (GraphPad^©^). Seropositivity was defined as a titer of ≥20.

### 2.4. T-Cell Response

The cellular response was randomly evaluated in study participants before and after the third vaccine dose. SARS-CoV-2 Spike-specific T-cell responses were assessed ex vivo. Heparinized whole-blood samples were stimulated with Spike protein and a control with no antigen (SARS-CoV-2 IGRA stimulation tube set, EuroImmun, Germany), strictly following the manufacturer’s protocol. After 24 h of stimulation, plasma was collected and secreted IFNγ was quantified (ELISA DuoSet, R&D Systems, Minneapolis, MN, USA). Results are presented as the difference between IFNγ levels in response to spike antigen vs. background response to no antigen control. Values > 50 pg/mL of Spike-specific response were considered positive.

### 2.5. Outcomes

The primary outcome was the proportion of seropositivity among LTRs one month following the third vaccine dose. Secondary outcomes included: the proportion of seropositive neutralizing antibody titers before and after the third vaccine dose, the proportion of patients with a positive cellular immune response before and after the third vaccine dose, serious adverse events following vaccination, and documented COVID-19 infection episodes. 

### 2.6. Statistical Analysis

Quantitative variables were expressed as means ± standard deviation for normally distributed variables and median (IQR) for those distributed abnormally. Categorical variables were expressed by frequencies. The univariate analysis compared baseline characteristics, continuous normally distributed variables were compared by independent *t*-test or ANOVA, and abnormally distributed variables were compared by Mann–Whitney test or Kruskal–Wallis as required. In addition, categorical variables were compared by the Chi-Square test (or Exact Fisher test, as appropriate). Multiple logistic regression was used for the multivariate analysis. Statistical significance was set at *p* < 0.05. Statistical analysis was performed using the SPSS (v27).

## 3. Results

### 3.1. Study Population

Of 180 LTRs vaccinated with two doses of the BNT162b2 vaccine, 139 patients received the third vaccine dose and were enrolled in the study. Overall, 137 LTRs, two heart and lung transplant recipients, and 41 healthcare workers (HCW) were included in the final analysis. The mean age was 56.60 ± 13.11 years, and 107 patients (59.8%) were males. Among LTRs, the median time between the second and third vaccine doses was 184 days (IQR 180–191), and the median time from transplant to the third vaccine dose was 44 months (IQR 19–96). The immunosuppressive regimen included: 132 patients (95%) who received tacrolimus, seven patients (5%) who received cyclosporine A, and 23 patients (16.5%) who received a regimen of low-dose CNI and everolimus. In addition, antimetabolite treatment included mycophenolate mofetil in 128 patients (92.1%). Mean immunosuppressive through levels were 9.48 ± 2.08.

### 3.2. Humoral Response

Overall, 59 patients (42.4%) had a seropositive antibody titer following the third vaccine dose (Figure 1).

The median S-IgG antibody concentrations before and after the third vaccine dose were 6.55 AU/mL (IQR 1.67–31.47) and 16.05 AU/mL (IQR 1.85–490.07), respectively. The median S-IgG antibody concentrations of the seropositive group were 34.80 AU/mL (IQR 9.10–209.35) ~3 months after the second vaccine dose and 713.40 AU/mL (IQR 174.80–2398.40) following the third (booster) vaccination (Table 1). As a comparison, all patients in the control group had a seropositive titer following the third vaccine dose, with S-IgG titers of 1072.55 AU/mL (IQR 681.67–2467.75) and 7807.30 AU/mL (IQR 3930.90–15461.80) before and after the third vaccine dose, respectively.

Among LTR participants, 29/109 (26.6%) patients with a negative titer following two doses have seroconverted following the third vaccine dose. Seropositivity was associated with younger age (t = 1.846, *p* = 0.033), a higher baseline GFR level (t = −2.800, *p*-value = 0.003), longer duration from lung transplantation (t = −1.992, *p* = 0.024) and a trend towards lower immunosuppressive drug levels 1–3 months before vaccination (H = 3.290, *p* = 0.072). No differences between patients with or without seropositive titer regarding primary lung disease and significant comorbidities, including DM (χ2 = 0.119, *p* = 0.730), IHD (χ2 = 2.123, *p* = 0.145) and CHF (fisher’s exact test, *p* = 0.401) were observed.

Ninety-seven percent of the seropositive group had a positive neutralizing antibody titer. We found a strong correlation between higher S-IgG levels and a positive neutralizing assay (r = 0.9551, *p* < 0.001, Figure 2).

Patients with neutralizing antibodies were younger (t = 2.047, *p* = 0.023) and were less likely to suffer from stage 3 or higher chronic kidney disease (Fisher Exact test, *p* = 0.035). We found a trend for positive neutralizing assay and higher baseline GFR (t = −1.667, *p* = 0.051). No difference was found between groups in the BMI (t = 1.135, *p* = 0.131), time from transplantation (U = 77.5, *p* = 0.371), or in the proportion of patients with CHF (Fisher Exact test, *p* = 1.000), or DM (Fisher Exact test, *p* = 0.613). In a multivariate analysis, seropositive antibodies were associated with lower BMI, a seropositive titer before the third vaccine dose, a longer duration from transplantation, and less severe chronic kidney disease. There was a trend toward a younger age and lower mean immunosuppressive drug trough levels (Table 2).

### 3.3. Cellular Response

Of 139 LTRs, the cellular response was evaluated in 86 patients before and 29 LTRs following the third vaccine dose. IFNγ-T-cell responses were detected in 9.4% following the second vaccine dose (unpublished data) and 17.2% after the third dose. Positive cellular response after the third vaccine dose was associated with a positive antibody titer following the second vaccine dose (Fisher’s Exact test, *p* = 0.022). Higher IFNγ-T-cell response was associated with a trend for lower BMI (U = 20.0, *p* = 0.051) and longer duration from transplantation (U = 28.5, *p* = 0.065). The IFNγ-T-cell response was not correlated with baseline GFR (U = 54.00, *p* = 0.729), the presence of chronic kidney disease beyond stage 3 (*p* = 1.000), IHD (*p* = 1.000), CHF (*p* = 1.000), or DM (*p* = 1.000).

### 3.4. Safety

No vaccine-related serious adverse events were observed during the study period. Pain at the injection site was reported in 60% of patients following the third vaccine dose. 

Two LTR participants, one with a seropositive titer and one with a seronegative titer, tested positive for COVID-19 infection during follow-up. None had severe disease, and both patients are alive to date.

## 4. Discussion

Our findings indicate that LTRs mount an improved immune response after receiving the third dose of the BNT162b2 vaccine, with overall seropositivity increasing from 18% to 42.4% and a seroconversion rate of 26.6%. Among previously seropositive patients, the median antibody titer increased by 2.5-fold, and 97% of seropositive LTRs developed neutralizing antibodies. This is the largest study reporting neutralizing antibody rates and cellular responses among vaccinated LTRs. 

Previous studies have shown that the third vaccine dose enhances immunogenicity in SOTRs [16,19] and LTRs [14], yet immunogenicity is still significantly lower compared to healthy immunocompetent individuals. Alejo et al. speculated that the attenuated immune response might be due to intensified immunosuppressive regimens [17]. Similar to our previous study [14], various previous studies have found mycophenolic acid-based regimens to be a risk factor for seronegative responses following the second vaccine dose [12,16,20,21,22]. In the current study, these results were not duplicated. Studies to evaluate the effect of immunosuppression reduction before booster dose administration among liver and kidney transplant recipients are ongoing [23,24]. Holding antimetabolites before the third dose vaccination increases antibody response levels in the antimetabolite-hold group, with stable DSA levels, FEV1, and no episodes of acute rejection. However, this was a small retrospective study, yet a brief hold of antimetabolites before booster dosing may be promising. Our current study is the first to examine the effect of immunosuppression drug trough levels on seropositivity. We found a trend towards higher immunosuppression trough levels among seronegative LTRs, which did not reach statistical significance. 

Additional variables associated with an attenuated immune response in our cohort included older age, a shorter duration from transplantation, and a lower baseline GFR. These data are consistent with previous studies in immunocompetent individuals and SOTRs [19,21,25,26,27,28,29]. In immunocompetent individuals, aging leads to a progressive reduction in immune system function, a phenomenon called immunosenescence. This phenomenon is thought to lead to the development of age-related diseases, reduced resistance to infection, and poorer responses to vaccination [22]. Reduced vaccination success is well documented in influenza and pneumococcal vaccines [28,29]. In COVID-19, the humoral response has been demonstrated to be age-dependent [23]. Specifically, reduced antibody and neutralizing antibody titers were significantly lower in the elderly [24,25]. Similar age-related reduced responses have been shown among SOTR [18,26]. In a large retrospective study including over 1000 LTRs, Dauriat et al. also found older age and the use of Mycophenolate to be associated with a negative serological response following the third vaccine dose [21]. However, their cohort showed a significantly lower seropositivity rate of only 16% following the third dose compared to our study. Previous reports regarding SOTRs, including LTRs, demonstrated higher seropositivity rates ranging from 55–68% following a third vaccine dose [30,31].

Longer time from transplantation was associated with an improved immunological response among our study participants in univariate and multivariate analyses. This aligns with previous studies and is assumed to be related to decreased immunosuppression intensity over time [21,32]. Of note, in our current clinical practice, we maintain our transplant recipients at relatively high CNI trough levels (as shown in Table 1) even far out from the transplant. This fact suggests that other immune factors may play a role in immune system responses besides immunosuppression intensity.

Chronic kidney disease (CKD) is a well-known risk factor for a reduced humoral response to many vaccines [33]. Our study supports prior results in SOTRs, in which seronegative recipients had a lower baseline GFR [19]. In kidney transplant recipients, Yahav et al. showed that a higher GFR contributed to positive seroconversion rates following the third vaccine dose [22]. Our study shows a correlation between advanced CKD (stage 3 and above) and a reduced level of antibodies after vaccination. This highlights the fact that impaired kidney function can lead to a weakened immune system response.

Our study found a strong correlation between positive S-IgG antibody titers and positive neutralizing antibody titers (r = 0.9551, *p* < 0.001). Ninety-seven percent of the seropositive LTRs demonstrated a positive neutralizing antibody titer. Our findings are consistent with a recent study in healthy adults aged 60 years or older, in whom positive neutralizing titers remained adequate three months after the third BNT162b2 vaccine and correlated with seropositive S-IgG titers [18]. Only a few studies have investigated neutralizing antibodies in SOTRs so far. Karaba and colleagues found that the third vaccine dose increased neutralizing antibodies among SOTRs, yet levels were far below that of healthy controls. Their study also demonstrated a strong correlation between anti-S IgG and neutralization; however, their cohort was small and included 47 SOTRs and only two LTRs [15]. As noted, not all antigen-specific antibodies have effective neutralization ability [34], which may explain why low S-IgG levels correlate with lower neutralizing antibodies. When we used a cutoff value of ≥100 to define seropositivity in the neutralization assay, as Evans et al. [35] suggested, only 68% of our cohort, particularly those with higher S-IgG titers, were positive. Currently, the cutoff values for S-IgG titers lack clarity and have not been definitively linked to clinical effectiveness. Identifying better cutoff thresholds may help establish protection against COVID-19, but this still needs further research, including clinical trials and real-world validation. Furthermore, although no clear correlates of protection were defined for COVID-19, the involvement of both humoral and cellular responses for protective immunity is well characterized [36,37]. The relationship between humoral and cellular response is of much interest in SOTRs. In a small study of 15 LTRs, Havlin et al. demonstrated a 47% cellular response rate among seronegative patients, suggesting that vaccination may offer an added clinical benefit regardless of seropositivity [38]. The current study’s strong correlation between cellular and humoral responses aligns with previous data following pneumococcal vaccination among kidney transplant recipients [39]. This strongly suggests that a clinically significant cellular response in seronegative patients is less likely, mainly in SOTRs in whom immunosuppression regimens are T-cell targeted. 

This study adds to prior data among SOTRs vaccinees, showing a constant increase in both antibody titer as well as the percentage of seropositivity rate following booster doses [19,22,29,31,40]. Most recently, a fourth vaccine dose in SOTRs significantly increased humoral response [41]. Preliminary data supported the use of monoclonal antibodies, both as pre-exposure prophylaxis and as treatment in immunocompromised patients such as SOTRs who did not elicit an adequate immune response [42,43,44]; however, constant virus mutations and new emerging variants have shown to escape monoclonal antibody activity, limiting their use due to inefficacy [45] and revoking their FDA emergency use authorizations [4,46,47]. Immune-evasion capabilities of new variants to monoclonal antibodies, alongside increased vaccine immunogenicity responses to repeated doses, underscores the significance of ongoing updated booster doses. The importance of vaccination among the vulnerable lung transplant population is intensified by repeated studies showing better outcomes for COVID-19 infection among vaccinated SOTRs, including LTRs, despite lower humoral responses detected by conventional methods [48,49,50,51]. 

Our study has some limitations. First, this is a single-center cohort analysis. Second, the t-cell immune response was evaluated only in a limited number of study participants. Last, the study was not designed to evaluate protective immunity against emerging COVID-19 variants but rather to assess LTRs immunogenicity. 

## 5. Conclusions

In conclusion, the current study adds to the growing body of evidence demonstrating that albeit LTRs elicit a weaker antibody response following COVID-19 vaccination compared to healthy individuals, the use of a booster dose results in the enhancement of both antibody response and the percentage of seropositive patients. Since LTRs are prone to severe COVID-19 morbidity and the emergence of new variants has limited monoclonal antibody efficacy, vaccination remains crucial for this vulnerable population. More research is needed to determine the clinical efficacy of vaccination, assess the utility of added booster doses, and evaluate proposed strategies to individualize immunosuppressive regimens in SOTRs to maximize vaccine effectiveness.

## Figures and Tables

**Figure 1 vaccines-11-00799-f001:**
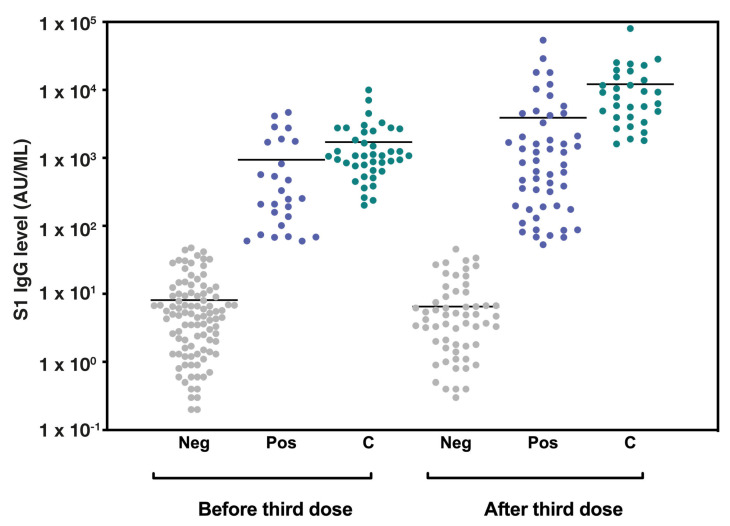
Seropositivity among LTRs and control group before and after the third vaccine dose. A column scatter plot graph representing the BNT162b2 vaccine-induced S1-IgG antibody levels (AU/mL) in lung transplant recipients and control group before and after the third vaccine dose. The bars represent median values. Abbreviations: LTRs, lung transplant recipients; AU, antibody units; Neg, LTRs with seronegative antibody titer: Pos, LTRs with seropositive antibody titer; C, control group antibody titer.

**Figure 2 vaccines-11-00799-f002:**
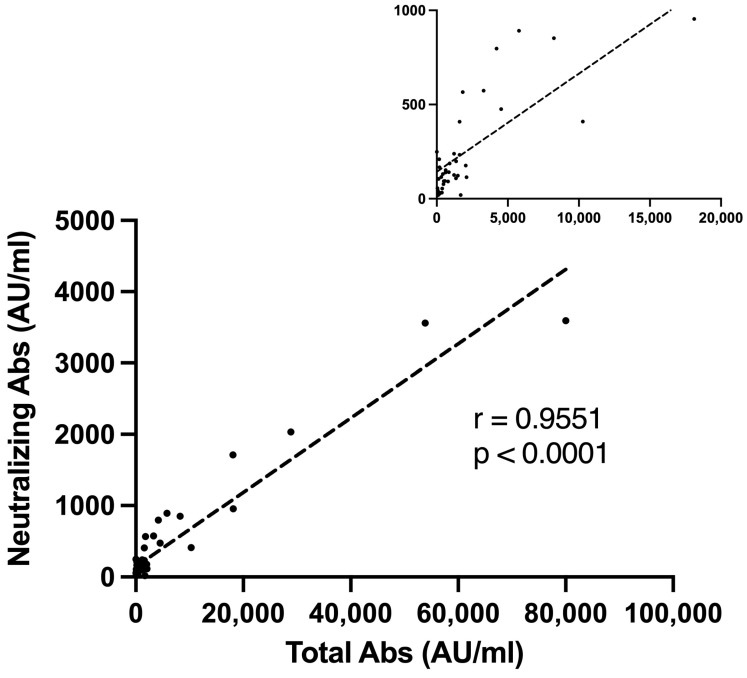
Neutralizing antibodies and seropositive s-IgG titer Correlation. A Pearson correlation plot representing the correlation between S1-IgG antibody levels (AU/mL) and neutralizing antibody titer among vaccinated lung transplant recipients. The small upper Pearson correlation plot represents a magnified view with a scale of 5000 Total Abs (AU/mL). Abs, antibodies; AU, antibody units.

**Table 1 vaccines-11-00799-t001:** Demographic and clinical characteristics of study participants, stratified by seropositivity to anti-spike IgG antibodies after three vaccine doses.

	Seropositive Group (n = 59)	Seronegative Group (n = 80)	*p* Value
S-IgG after the third vaccine dose (AU/mL, (median, IQR)	713.40 (174.80–2398.40)	3.25 (0.40–6.42)	**<0.001**
S-IgG before third vaccine dose (AU/mL) (median, IQR)	34.80 (9.10–209.35)	2.55 (0.77–6.52)	**<0.001**
Age (mean, SD)	54.03 ± 13.57	57.86 ± 12.75	**0.033**
Male gender (number, percentage)	37 (62.7%)	70 (58.3%)	0.574
BMI (per kg/m2) (mean, SD)	26.39 ± 4.89	26.14 ± 4.44	0.740
GFR (per ml/min/1.73m^2^) (mean, SD)	69.58 ± 23.06	58.51 ± 23.00	**0.003**
Chronic kidney disease (stage 3 and above) (number, percentage)	7 (11.9%)	25 (31.3%)	**0.007**
Time from transplantation (months) (mean, SD)	85.22 ± 63.83	65.25 ± 54.10	**0.024**
Immunosuppression regimen (number, percentage)			
Mycophenolate mofetil	54 (91.5%)	73 (92.4%)	1.000
CNI’s	59 (100%)	80 (100.0%)	1.000
mTOR inhibitor	9 (15.3%)	14 (17.5%)	0.725
Immunosuppression drug through levels (IU) (mean, SD)	9.08 ± 2.07	9.80 ± 2.04	0.072

Significant *p* values (*p* < 0.05) are in bold. Abbreviations: BMI, body mass index; GFR, glomerular filtration rate; CNIs, calcineurin inhibitors; mTOR, mammalian target of rapamycin.

**Table 2 vaccines-11-00799-t002:** Multivariate analysis for seropositivity following the third vaccine dose.

	Confidence Interval	OR	*p* Value
BMI	1.049–1397	1.211	**0.009**
Severe CKD (stage 3 and above)	0.022–0.471	0.101	**0.004**
Time from transplantation	1.000–1.020	1.010	**0.048**
Baseline seropositivity	7.070–234.709	40.736	**<0.001**
Immunosuppression drug trough levels	0.591–1.034	0.782	0.085
Age	0.915–1.006	0.960	0.088

Significant *p* values (*p* < 0.05) are in bold. Abbreviations: OR, odds ratio; BMI, body mass index; CKD, chronic kidney disease.

## Data Availability

Data available on request due to restrictions eg privacy or ethical.

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
