# Peer review of "Immunogenicity of a Third Dose of BNT162b2 Vaccine among Lung Transplant Recipients—A Prospective Cohort Study"

_vaccines, 2023, doi:10.3390/vaccines11040799_

Round 1

Reviewer 1 Report

This work focused on monoclonal antibodies that have limited effetiveness against prevalent sub variants and LTRs to severe COVID-19 morbidity and the importance of vaccination.

The authors need to add clear research question and clear research hypothesis in their introduction.

Also, explain what do you mean by The cellular response was evaluated in 86 and 29 LTRs before and after the third vaccine dose.

in discussion, please refer to some previous published studies.

little english spelling errors

Author Response

Dear Reviewer,

We would first like to thank you for the opportunity to submit a revised version of our manuscript. We would also like to thank you for your in-depth and insightful comments and questions, improving our manuscript. We have accepted all comments and revised the text accordingly.

  1. Point 1: The authors need to add a clear research question and clear research hypothesis in their introduction.

Response 1: Thank you for drawing our attention to this important point. As suggested, we added our hypothesis to the text and adjusted the last paragraph of our introduction to quote: “Given the heightened risk for severe COVID-19 disease among LTRs and data showing viral immune escape from monoclonal antibodies, our study aimed to evaluate the immunogenicity humoral and cellular response of a third dose of the BNT162b2 vaccine among LTRs. We hypothesized that similar to data following the second dose we would observe an increase in immunogenicity among LTRs following a third vaccine dose. 

  1. Point 2: Also, explain what you mean by The cellular response was evaluated in 86 and 29 LTRs before and after the third vaccine dose.

Response 2: Thank you for this comment. Of note, we could not evaluate cellular responses in the entire cohort, only a sample of patients, due to logistical difficulties. We agree with the reviewer that this point needs further explanation. Therefore, to clarify, we added to the text: “of 139 LTRs, the cellular response was evaluated in 86 patients before and 29 LTRs following the third vaccine dose”.

  1. Point 3: In discussion, please refer to some previous published studies.

Response 3: Thank you for this comment. We added further references which we may not have included in the first version. We have focused our discussion on previous studies published in SOTR, such as references 12, 13, 15, 16, 19–23, 29, 30, and specifically in LTRs such as 14,17, 20-21, 36,38.  

  1. Point 4: little English spelling errors

Response 4: Thank you for pointing this out. we have reviewed the entire manuscript again and corrected all spelling errors we encountered.

We would again like to thank you for the time and effort invested in this review.

Yours sincerely,

Yael Shostak, MD Deputy Director, Department of Internal Medicine D, Rabin Medical Center, Beillison Hospital, Petach Tikva, Israel

Reviewer 2 Report

Thank you for inviting me to review this manuscript. Shostak et al. evaluated the humoral response and neutralizing antibodies against SARS-CoV-2 S antigen in 139 vaccinated lung transplant recipients almost 4-6 weeks after the third mRNA vaccine dose and was compared to 41 healthcare workers. Furthermore, cellular immunity was evaluated in a proportion of participants.

More than 42% of the lung transplant recipients had a seropositive antibody titer, and 17% had a positive t-cell response. The authors further determined some of the factors that were associated with seropositivity.

The subject is interesting and important, and the manuscript is well-written. I have no further comments.

Author Response

Dear Reviewer,

We would like to thank you for the opportunity to submit a revised version of our manuscript. We would also like to thank you for your kind remarks, and the time and effort invested in this review.

Yours sincerely,

Yael Shostak, MD Deputy Director, Department of Internal Medicine D, Rabin Medical Center, Beillison Hospital, Petach Tikva, Israel

Reviewer 3 Report

Following their previous study, Shostak et al conducted a comprehensive analysis of the humoral and cellular responses after the third dose of the BNT162b2 vaccine among lung transplant patients. They observed enhanced immune responses post-3rd vaccination. Overall, the study design is solid and the manuscript is well-written. It still could be improved with some minor revisions.

1. please improve the resolution of the figures. The scale of figure 2 looks inappropriate. 

2. Would be worth discussing the different observations in the present study and what was shown by Gaëlle Dauriat et al.  

Author Response

Dear Reviewer,

We would first like to thank you for the opportunity to submit a revised version of our manuscript. We would also like to thank you for your in-depth and insightful comments and questions, improving our manuscript. We have accepted all comments and revised the text accordingly.

  1. Point 1: Please improve the resolution of the figures. The scale of figure 2 looks inappropriate.

Response 1: thank you for this important comment. We have adjusted the resolution of the figures and edited the scale appropriately in figure 2. In addition, we reloaded the new figures to the editorial site. You may find adjusted Figure 2 attached.

  1. Point 2: Would be worth discussing the different observations in the present study and what was shown by Gaëlle Dauriat et al.  

Response 2: Thank you for this important remark. As advised, we have further discussed the differences in observations between the two studies as we added the following text: “In a large retrospective study including over 1000 LTRs, Dauriat et al. also found older age and the use of Mycophenolate to be associated with a negative serological response following the third vaccine dose [20]. However, their cohort showed a significantly lower seropositivity rate of only 16% following the third vaccine dose compared to our study. Previous reports regarding SOTRs, including LTRs, demonstrated higher seropositivity rates ranging from 55-68 % following a third vaccine dose.”

We would again like to thank you for the time and effort invested in this review.

Yours sincerely,

Yael Shostak, MD Deputy Director, Department of Internal Medicine D, Rabin Medical Center, Beillison Hospital, Petach Tikva, Israel

Reviewer 4 Report

Thank you for sharing this interesting article. Here some very minor comments that may help to improve the article:

L74: Please explain in your manuscript the vaccination schedule for the 3 doses of BNT162b2. How was informed consent obtained? 

L80/82: Please be more specific in your manuscript regarding the blood sample collection. Was a whole blood sample drawn each 4-6 weeks and 3 months after the 3rd vaccine dose? It is not fully clear in the manuscript, please revise accordingly. 

Author Response

Dear Reviewer,

We would first like to thank you for the opportunity to submit a revised version of our manuscript. We would also like to you for your in-depth and insightful comments and questions, improving our manuscript. We have accepted all the comments and revised the text accordingly.

Point 1: Please explain in your manuscript the vaccination schedule for the 3 doses of BNT162b2.

Response 1: Thank you for your comment. As advised, we added the vaccine schedule to the text: “Patients were vaccinated as advised by the Israel Ministry of health at three points: the first dose between December 2020 and January 2021, and the second dose was administrated three weeks following the first dose. The third dose was available for eligible at-risk populations as advised by the Israel Ministry of health between July 2021 and August 2021”.

  1. Point 2: How was informed consent obtained? 

Response 2: Thank you for pointing out this important issue. In line 80 we clarified using the sentence: “All participants provided written informed consent.” 

  1. Point 3: L80/82: Please be more specific in your manuscript regarding the blood sample collection. Was a whole blood sample drawn each 4-6 weeks and 3 months after the 3rd vaccine dose? It is not fully clear in the manuscript, please revise accordingly.

Response 3: We agree that this point needs to be thoroughly understood and requires rephrasing. To clarify, we rephrased the paragraph and added the following information within the text: “LTRs were evaluated, and whole blood samples were drawn for antibody titers, leukocyte and lymphocyte count, creatinine level, and immunosuppressive drug trough levels (calcineurin and mTOR inhibitors), 4-6 weeks and then again three months 10-15 weeks following the third vaccine dose. These were compared to baseline antibody titers drawn 9-18 weeks before the third vaccine dose. Whole blood samples for antibody titer in the control group were drawn nine weeks before and 3-5 months following the third vaccine dose”.

We would again like to thank you for the time and effort invested in this review.

Yours sincerely,

Yael Shostak, MD Deputy Director, Department of Internal Medicine D, Rabin Medical Center, Beillison Hospital, Petach Tikva, Israel